# A Double-Blind Randomized Placebo-Controlled Study Assessing the Safety, Tolerability and Efficacy of a Herbal Medicine Containing Pycnogenol Combined with Papain and *Aloe vera* in the Prevention and Management of Pre-Diabetes

**DOI:** 10.3390/medicines7040022

**Published:** 2020-04-22

**Authors:** Luis Vitetta, Belinda Butcher, Serena Dal Forno, Gemma Vitetta, Tessa Nikov, Sean Hall, Elizabeth Steels

**Affiliations:** 1Sydney Medical School, Faculty of Medicine and Health, The University of Sydney, Sydney NSW 2006, Australia; drbeth@evidencesciences.com.au; 2Medlab Clinical, Sydney NSW 2015, Australia; serena_dalforno@medlab.co (S.D.F.); gemmavitetta@gmail.com (G.V.); t.nikov@unsw.edu.au (T.N.); sean_hall@medlab.co (S.H.); 3WriteSource Medical Pty Ltd., Lane Cove NSW 2066, Australia; bbutcher@writesourcemedical.com.au; 4School of Medical Science, University of New South Wales (UNSW), Sydney NSW 2052, Australia; 5Facility of Science, Health, Education and Engineering, University of the Sunshine Coast, Sippy Downs QLD 4556, Australia

**Keywords:** pre-diabetes, pycnogenol, papain, *Aloe vera*, impaired fasting glucose, fasting plasma glucose

## Abstract

**Background:** Herbal medicines present attractive options to patients with chronic diseases. Undertaking clinical studies with patients presenting with symptomless pre-T2D can lead to significant limitations. **Methods:** A 12-week randomized double-blind placebo-controlled clinical study was conducted that investigated the safety and efficacy of an herbal formulation administered orally for the treatment of pre-type 2 diabetes (pre-T2D). **Results:** A numerically greater proportion of subjects in the interventional arm had impaired fasting glucose (IFG) at week 12 compared to the control arm (71.0% vs. 69.0%, p = 0.75). Fewer participants had impaired glucose tolerance (IGT) at 12 weeks in the intervention arm compared to the control arm (unadjusted 58.3% vs. 66.7%, p = 0.65; adjusting for baseline IGT, p = 0.266). In a subgroup analysis, subjects with a baseline fasting plasma glucose (FPG) level in the range of 6.1–6.9 mmol/L demonstrated a non-significant lower proportion of IFG at week 12 in the intervention arm compared to the control arm (60.0% vs. 41.7% p = 0.343). Total blood cholesterol and triglyceride levels remained unchanged from baseline to week 12 in both treatment groups. **Conclusions:** This study suggests that a polyherbal medicine was not effective for reducing the metabolic markers associated with pre-T2D over a 12-week period. Therefore, larger studies with well-defined endpoints and of longer duration are warranted.

## 1. Introduction

Pre-diabetes is defined as a condition in which fasting plasma glucose (FPG) levels are elevated above the normal range but do not satisfy the criteria for the diagnosis of diabetes mellitus [1,2]. A single FPG level measurement of between 6.1 and 6.9 mmol/L defines impaired fasting glucose (IFG) and is used to diagnose pre-diabetes [3]. However, FPG levels in the range of 5.5–6.0 mmol/L do indicate an increased risk of developing type 2 diabetes (T2D) [4], especially if linked to adverse lifestyle outcomes such as being significantly overweight.

In a cross-sectional survey of adults aged 25 years and older, impaired glucose tolerance (IGT) is when blood glucose levels are higher than normal but not high enough to be classified as T2D. Impaired glucose tolerance was present in 10.6% of subjects, being more common in women (11.9% vs. 9.2% in men). IFG was present in 5.8% of subjects, being more prevalent in men (8.1% vs. 3.4% in women). This represents an overall pre-diabetes prevalence of 16.4% in Australians who are older than 25 years [3,5,6]. Pre-diabetes precedes the development of T2D and is linked to relative insulin deficiency and tissue insulin resistance, which cause abnormal blood glucose levels [5]. Risk factors for pre-diabetes include being overweight/obese, increased waist circumference and increasing age. People who develop pre-diabetes are at increased risk of developing diabetes, cardiovascular disease and other macrovascular diseases, with approximately 3% to 10% of patients with pre-diabetes going on to develop diabetes [1,2].

Overall, pre-diabetes confers an approximate six-fold increased risk of diabetes compared with patients who have normal glucose tolerance. The management of pre-diabetes is determined by the increased risk of developing both diabetes and cardiovascular disease [7]. First-line treatment involves lifestyle intervention (diet and exercise). Patients may require metformin, which produces the greatest benefit in patients aged below 60 years and in those who are overweight. A recent meta-analysis demonstrated that lifestyle interventions reduced progression to diabetes equal to that of pharmacological interventions and, while patient adherence to diet and exercise interventions was high in the studies assessed, it is unknown whether adherence continues beyond that of the trial period [8].

In 2000, an estimated 171 million people worldwide were diagnosed with T2D and that number is projected to double by 2030. Interventions to prevent T2D are, therefore, an important area of research for future health policy. Given that people with IGT are at greater risk of developing T2D, improving blood glucose control and insulin sensitivity is paramount in such patients [8].

Pycnogenol is a standardized extract from the French maritime pine bark (*Pinus pinaster*) and consists of phenolic compounds. Pycnogenol has demonstrated clinical efficacy in lowering blood esglucose levels (FPG, HbA1c and endothelial-1) [9] and improving cardiovascular risk factors (blood pressure, plasma endothelial-1, HbA1c, FPG, low-density lipoprotein and urinary albumin) [10] in patients with T2D. Pycnogenol also improves metabolic syndrome risk factors (FPG, waist circumference, plasma triglycerides, blood pressure and increased high-density lipoproteins) [11]. Furthermore, pine bark extract can reduce FPG in people diagnosed with diabetes, blood pressure in mild to moderate hypertensive patients, and waist circumference as well as improve lipid profile, renal and endothelial functions in metabolic syndrome [12]. The formulation investigated also included papaya (*Carica papaya* L.), which has health benefits that have been reported to include the control of hypertension, diabetes and obesity [13,14]. Similarly, the inclusion of *A. vera* L. Burm.f. has also been reported to possess several beneficial metabolic properties, such as lipid-lowering, anti-hypertensive, anti-diabetic, anti-obesity, and cardioprotective effects [15], that could potentially augment the effects of pycnogenol and papaya.

The purpose of this study was to assess the efficacy of a pycnogenol formulation combined with papain, an anti-inflammatory enzyme derived from *C. papaya* L., and *A. vera* L. Burm.f. (i) improving impaired FPG levels and IGT in patients presenting with pre-T2D who may have been prescribed metformin and in those not on any pharmacological interventions; (ii) in subjects exhibiting the risk factors but who have yet to progress to a diagnosis of pre-T2D, such as being overweight, lacking in physical activity or having a family history of diabetes.

## 2. Materials and Methods

A randomized, double-blind, placebo-controlled study was conducted with 117 participants who had an FPG level ranging from 5.5 to 6.9 mmol/L and a body mass index of 25 kg/m^2^ or more. Prospective participants had to be willing to not change their dietary or physical activity habits. On inclusion, participants were randomized to receive a 12 week treatment with either 50 mL b.i.d. of the dietary supplement or matched placebo, consisting of 3 mg of sodium chloride plus excipients [potassium sorbate, sodium benzoate, honey, acetate, wildberry flavour, and anthocyanins] in equivalent proportions as in the active formulation. The supplement consisted of 2.6 mg/mL of pycnogenol from the French maritime pinebark extract, 2.4 mg/mL of papain 1.75 mg/mL of *A. Vera* and 3 mg of sodium chloride/mL plus excipients, contained in a vial at a final volume of 50 mL. Patients were followed up until 12 weeks post randomization. This clinical study was approved on 16 February 2015 (HREC 00252) by the National Institute of Integrative Medicine’s Human Research Ethics Committee (NHMRC registered EC00436), TGA CTN registered number 2015/0180 and Australian Clinical Trail Registry number ACTRN12615000233527.

### 2.1. Participants

Participants were enrolled between March 2015 and April 2018. The last participant completed this study in July 2018. Eligible participants were 18 years of age or older at time of entry into study; males and females with impaired fasting blood glucose (fasting blood glucose 6.1–6.9 mmol/L); had an initial Australian Type 2 Diabetes Risk Assessment Tool (AUSDRISK) questionnaire score of six or more; a body mass index of 25 or over; a waist measurement of 102 cm or over in men or 88 cm or over in women; had the ability to understand the informed consent process and to give informed consent to the experimental treatment; agreed to undergo multiple venipunctures; agreed to adhere to the study protocol, including not changing diet or exercise patterns over the 12 week study period. Participants signed a written informed consent form.

A protocol HREC amendment was submitted to include subjects with a fasting blood glucose of 5.5 to 6.1 mmol/L.

### 2.2. Design of This Study

This study was a randomized, single-site, double-blind, placebo-controlled, parallel-arm group study conducted at the Medlab Clinical facility in Alexandria, Sydney (NSW). A computer-generated random-block size procedure was used to randomize participants in a 1:1 fashion to receive either the intervention or placebo. The randomization code was generated by the independent statistician (BB). Blinded randomization codes were held by the independent statistician and participants were allocated to ‘A’ or ‘B’ by contacting the statistician. The unblinding information was held by an independent researcher within Medlab Clinical. This study was conducted according to the ethical principles of the Declaration of Helsinki, ICH-GCP and local regulations. All clinical trial registrations and approvals are presented in Table 1.

Eligible participants underwent a baseline pathology test at their local Pathology Centre (Laverty or other centre). Baseline testing included an oral glucose tolerance test (OGTT), insulin, high-sensitivity C-reactive protein (hs-CRP), HbA1c, FBC, ELFTs, urea and a lipid profile. Following initial laboratory screening, eligible participants attended the Medlab Clinical facility, where prospective participants documented all demographic data, anthropometric measurements (blood pressure, waist–hip circumference, weight and height). Participants completed the quality of life questionnaire (IWQOL-Lite), a 3 day diet recall, and a physical activity (L-Cat) questionnaire. Eligible participants were then randomized and provided with sufficient study medication or placebo for a period of 4 weeks. Subjects were provided with instructions on how to administer the test medication or placebo and a study diary to record daily intake of the administered vials ‘A’ or ‘B’. Participants underwent FPG testing at 4, 8 and 12 weeks. At the 12 week time point, follow-up pathology assessments for OGTT (+insulin), HbA1c, FBC, ELFTs, urea and lipid profile were performed at local pathology collection centres. Participants were required to visit the Medlab Clinical facility for a total of four study visits (not including FBG eligibility assessment) at 4 weekly intervals. At each visit, study diaries were collected, and further test vials labelled ‘A’ or ‘B’ and diaries were provided.

### 2.3. Outcome Measures

The primary objective of this study was to determine the effect of a herbal formulation containing pycnogenol, papain and *A. vera* as the active components in the test vial on the proportion of patients with IFG at week 12 compared to those treated with a placebo. The dose administered was 50 mL, b.i.d., containing pycnogenol (130 mg), papain (120 mg) and *A. vera* (87.5 mg) for 12-weeks.

The primary efficacy outcome measure was a change in FPG level at 12 weeks between groups administered intervention compared to placebo.

Secondary objectives included the effect of a herbal formulation containing pycnogenol, papain and *A. vera* as the principle actives in the supplement compared to placebo on: (i) the proportion of patients with IGT; (ii) hs-CRP level (iii) changes in HbA1c; (iv) safety (liver function, urea and full blood count); (v) HDL, LDL, TGs and lipid profile; vi) quality of life (SF12v2 questionnaire).

Secondary outcome measures were considered as exploratory and included changes at 12 weeks in other parameters of glucose homeostasis (including fasting plasma insulin, plasma glycated hemoglobin (HbA1c), and markers of insulin resistance/sensitivity), parameters of lipid homeostasis (plasma triglycerides, total cholesterol, high-density (HDL) and low-density (LDL) lipoprotein cholesterol, free fatty acids), adiposity markers (body weight, body mass index, fat mass and fat-free mass, adipocyte diameter), and markers of inflammation (high-sensitivity C-reactive protein (hs-CRP).

Tertiary objectives included the effect of a herbal formulation containing pycnogenol, papain and *A. vera* as the principle actives in the supplement compared to placebo on: (i) blood pressure; (ii) waist to hip ratio; (iii) weight; (iv) dietary habits (3 day diet recall); (v) physical activity (International Physical Activity Questionnaire).

### 2.4. Power and Sample Calculation

This clinical trial was designed to demonstrate the superior effect of pycnogenol, papain and *A. vera* over placebo on IFG at week 12 (Table 1). The primary endpoint was the proportion of patients with IFG at week 12. The null hypothesis was that there was no difference in the proportion of patients with IFG between the control and intervention arms. Initially, all patients had IFG, defined as an FPG level between 5.5 and 6.9 mmol/L (as part of the inclusion criteria for this study). It was hypothesized that there would be no improvement in the proportion of patients with IFG in the control arm, and that 10% of patients in the intervention arm will no longer be considered to have IFG (that is, an improvement of 10%). Using a one-sided alpha set at 0.05, the required number of eligible participants was 58 per group. With an anticipated drop out of up to 20%, this number was inflated to 73 patients per group.

### 2.5. Statistical Analysis

The primary analysis was based on the full analysis set (all randomized participants). No imputation for missing data occurred. A sensitivity analysis using per-protocol principles only included patients who had adhered to the study interval between visits (±7 days) and who had adhered to at least 80% of the study treatment (these participants were considered compliers).

All data were summarized descriptively using n, the mean ± standard deviation, or the median (25%–75% IQR), and frequency and percent for categorical data. Unadjusted comparisons between arms in the proportion of patients with IFG or IGT at baseline and week 12 were made using Chi-square tests. Logistic regression was used to determine whether there were differences between arms in the proportion of patients with IFG or IGT at week 12 including baseline IFG/IGT, treatment and gender as covariates in the model. Between-group comparisons of hs-CRP, HbA1c, HDL, LDL, TGs, LFT, urea, FBC, BP, waist to hip ratio, and weight will be conducted using ANOVA, with sex and baseline levels as covariates. Changes in quality of life were assessed according to the SF12v2 Scoring Manual.

Similarly, as previously reported [16] with a polyherbal supplement, this study in order to classify participants according to their capacity to secrete insulin and the estimated impact on FPG, the correlation between changes in FPG level and changes in insulin secretion (HOMA-B%) was tested between groups. The baseline values of the identified two groups of participants were compared using the non-parametric Mann–Whitney test to identify the two different phenotypes.

No adjustment for multiple comparisons was made. A statistical assessment was carried out on the full analysis set (FAS) of the population with n = 117 participants; and a per-protocol (PP) population analysis with n = 110 participants (Figure 1). All tests were conducted two-sided and *p* values of less than 0.05 were considered statistically significant. Statistical analyses were performed using Stata MP V15.2 for Mac (StataCorp, College Station, TX, USA).

## 3. Results

Participants were screened and recruited according to the CONSORT flow diagram presented in Figure 1. Of the 117 enrolled participants, 7 participants were found to have not met the inclusion and/or the exclusion criteria. These patients are included in the full analysis set (FAS) but excluded from the per-protocol (PP) analysis set.

Demographic and other baseline characteristics of subjects included in this study are presented in Table 1.

### 3.1. Efficacy Evaluation

#### 3.1.1. Impaired Fasting Glucose (IFG) (Primary Endpoint)

The primary efficacy outcome was the proportion of patients who had IFG at week 12 (Table 2). In this study, this was defined as an FPG level between 5.5 and 6.9 mmol/L.

A logistic regression analysis was performed on the screened population to determine the effect of the baseline IFG status, treatment and sex on IFG status at 12 weeks. In this adjusted analysis, baseline IFG status significantly affected IFG status at 12 weeks (*p* = 0.002) but treatment (*p* = 0.68) and sex (*p* = 0.51) did not. In the subset of patients who met the inclusion/exclusion criteria, baseline IFG status significantly affected IFG status at 12 weeks (*p* = 0.008), but treatment (*p* = 0.43) and sex (*p* = 0.65) did not.

There were 17 participants who were taking metformin at study baseline—seven in the intervention arm and 10 in the control arm (Table 1). All subjects were diagnosed at some previous time point prior to study commencement with IFG. A sub-analysis showed that at the end of week 12, there were no statistically significant differences in FPG levels prior to the OGTT or at the 1 h OGTT and 2 h OGTT (Figure 2).

#### 3.1.2. Impaired Glucose Tolerance (Secondary Endpoint)

The majority of patients did not have IGT at baseline, which was defined as an FPG level between 5.5 and 6.9 mmol/L (Table 3).

A logistic regression analysis was performed to determine the effect of baseline IGT status, treatment and sex on IGT status at 12 weeks. Baseline IGT status was significant (*p* < 0.001) in the model, but treatment (*p* = 0.72) and sex (*p* = 0.59) were not. Additional secondary endpoints are presented in Table 4.

### 3.2. Safety and Tolerability

There were no changes in the proportion of patients with abnormal blood test results from baseline to the end of the study at week 12. Furthermore, there were no reports of mild, moderate or serious adverse events during this study.

### 3.3. Quality of Life

No significant changes in quality of life occurred between treatment arms or from baseline to week 12 (Table 5).

### 3.4. Tertiary Endpoints

There were observed and recorded modest reductions in diastolic blood pressure in both groups, and these were greater in the control arm (−1.2 (13.6) vs. −5.6 (9.8), *p* = 0.08). These are unlikely to be clinically relevant changes.

### 3.5. Physical Activity

Physical activity scores were not significantly different by treatment allocation in the screened population at baseline and these were 2.8 (0.1) and 2.9 (1.1) for test versus placebo, respectively (*p* = 0.61). There were no significant changes in physical activity scores post completion of this study at 12 weeks between treatment and placebo groups—3.0 (1.2) and 3.0 (1.0), respectively (*p* > 0.99). The changes in physical activity scores for the treatment and placebo groups were 0.2 (1.1) and 0.1 (0.6) respectively at 12-weeks (*p* = 0.48).

### 3.6. Additional Physiological Measures

There were no differences in BMI, baseline FPG, OGTT, or insulin levels between the treatment arms (Table 6).

A sub-analysis of primary endpoints for subjects participating in this study with a baseline FPG level in the range of 6.1–6.9 mmol/L is presented in Table 7. A greater proportion of the participants randomized to the intervention arm did not have IFG at week 12 compared with those allocated to the control arm. While a trend was noted, it was not statistically significant (60.0% vs. 41.7% *p* = 0.343). It should be noted though that this comparison was not sufficiently powered (Power = 15%).

Moreover, there was also a trend for fewer participants to have IGT at 12-weeks in the intervention arm compared to the control arm (58.3% vs. 66.7%, *p* = 0.65)—when adjusting for baseline IGT, the p-value for treatment was 0.266. Again, this study was not sufficiently powered for this comparison.

Further analysis showed that there was a trend for participants allocated to the intervention arm at 12-weeks to have an FPG level that was lower than in participants allocated to the control arm. However, the trend was not statistically significant (*p* = 0.16). Further, at 12-weeks, fasting insulin levels were 7.5 mU/L higher in the placebo group compared to the treatment group and this trend was statistically significant (*p* = 0.039).

### 3.7. Blood Lipid Levels

There were no statistically significant differences by arm in either total cholesterol or triglyceride levels at baseline or at week 12 (Figure 3 and Figure 4).

Total plasma cholesterol had a medium (IQR) level for the placebo group, at a baseline of 4.9 mmol/L (4.4–5.5) and 5.3 mmol/L (4.5–6.0) at 12 weeks respectively (Figure 3); whereas total plasma cholesterol at baseline was 5.1 mmol/L (4.4–5.9) and 4.95 mmol/L (4.1–6.2) at 12-weeks for the treatment group respectively (Figure 3).

Total plasma triglycerides had a medium (IQR) level for the placebo group, at a baseline of 1.4 mmol/L (1.1–1.8) and 1.5 mmol/L (1.1–1.9) at 12 weeks respectively (Figure 4), whereas total plasma triglycerides at baseline was 1.6 mmol/L (1.2–2.2) and 1.7 mmol/L (1.1–2.2) at 12 weeks for the treatment group respectively (Figure 4).

## 4. Discussion

This clinical study demonstrated that a polyherbal formulation consisting of an extract of pycnogenol, papain and *A. vera* was not effective for reducing the metabolic/glycemic markers associated with a diagnosis of pre-T2D over a 12 weeks period. A numerically greater proportion of subjects in the interventional arm had impaired fasting glucose (IFG) at week 12 compared to the control arm.

Further analysis showed that there was a non-significant trend for participants allocated to the pycnogenol, papain and *A. vera* intervention arm to have lower FPG levels at 12 weeks than those in the control arm (*p* = 0.16). Further, at 12 weeks, fasting insulin levels were 7.5 mU/L higher in the placebo group compared to the treatment group and this trend was borderline significant (*p* = 0.039). Notwithstanding, a subgroup analysis of participants with a baseline FPG level in the range of 6.1–6.9 mmol/L reported a trend of fewer patients that had non-significant decreased IGT at 12 weeks in the polyherbal intervention arm compared to the control arm (58.3% vs. 66.7%, *p* = 0.65).

A logistic regression analysis was performed on the screened population to determine what effect the baseline IFG status, treatment allocation and gender would have on IFG status at 12 weeks. In this adjusted analysis, baseline IFG status significantly affected IFG status at 12 weeks (*p* = 0.002) but treatment allocation (*p* = 0.68) and gender (*p* = 0.51) did not. In the subset of patients who met the inclusion/exclusion criteria, baseline IFG status significantly affected IFG status at 12 weeks (*p* = 0.008, but treatment allocation (*p* = 0.43) and gender (*p* = 0.65) did not.

There were 17 participants who were taking metformin at study baseline—seven in the intervention arm and 10 in the control arm. These participants were diagnosed at some previous time point prior to study commencement with IFG. A sub-analysis showed that at the end of the study, there were no statistically significant differences in FPG levels between the test and control groups.

In concordance with our findings, a recent review [17] concluded that pine bark extracts such as pycnogenol are safe. In addition, reports [15,17,18] suggest that there is potential for the administration of pycnogenol, papain and *A. vera* in the treatment of inflammatory metabolic diseases such as diabetes.

Pre-diabetes is still considered as high risk for developing T2D, with a conservative estimated reported yearly conversion rate of between 5% and 10% [19].

Others report that transition to overt diabetes may be as high as 38% [20]. Notwithstanding, most patients diagnosed with metabolic abnormalities (e.g., obesity, dyslipidemia) will have a high probability of future progression to T2D [21]. Furthermore, individuals suspected to be pre-diabetic will have a significantly increased risk of developing associated T2D pathologies, such as diabetic neuropathy, retinopathy, macrovascular complications and nephropathy [21]. Blood lipid pathology tests for total plasma cholesterol and triglycerides showed no significant differences between treatment allocation groups at the end of the study—a trend that is probably mostly linked to levels of physical activity remaining unchanged throughout this study in both groups.

The initial hypothesis from the current clinical study was that all subjects inducted into this clinical trial would initially have IGT, defined as an FPG level between 6.1 and 6.9 mmol/L—given the current understanding of pre-T2D transition to T2D, this range was considered as an early form of T2D in this study [22]. As such, an HREC amendment was submitted requesting a protocol amendment that allowed subjects to be included in this study who presented with a lower level of IGT of 5.5 mmol/L.

We contend that patients diagnosed with pre-T2D present with an intermediate state of hyperglycemia, with glycemic parameters above normal but below the threshold for a diagnosis of T2D. This then fashions a further complication when conducting clinical studies with pre-T2D subjects in that the diagnostic criteria for pre-T2D are not constant across various international professional organizations (i.e., the World Health Organization, the International Diabetes Federation, and ADA), with differences of opinion as to what constitutes the appropriate criteria required to diagnose pre-diabetes. In the 10 year follow-up Chennai urban rural epidemiology study, investigating different cut off points for IFG on the incidence of T2D, Anjana and colleagues (2015) reported that the various categories of dysglycemia (i.e., low and high IFG) were associated with an increased risk of progression to T2D.

Microbial communities in the intestines have been reported through metagenome-wide association studies, presenting metagenomic data and clinical features showing that significant differences exist on the metagenomic level between metabolically healthy versus metabolically unhealthy individuals [23]. It is interesting to note that herbal extracts may positively impact the intestinal microbiome by assisting in managing metabolic syndrome symptoms [24]. This is in accordance with reports that show that pycnogenol extracts can be metabolized by the intestinal microbiota to produce compounds beneficial to health [25]. Furthermore, the anti-diabetic drug metformin has been reported to beneficially modulate the intestinal microbiome, with positive effects on glycemic control in T2D. Interestingly, a subgroup analysis aimed at establishing whether the polyherbal formulation provided an additive therapeutic effect when combined with metformin showed no significant differences.

While this clinical study does not support the use of a polyherbal formulation to support IFG over a 12 week period, there is the possibility that the polyherbal formulation may improve IFG levels over a longer period such as 24 weeks when combined with lifestyle changes and such a study would need to be adequately powered. As such, the impact of pycnogenol, papain and *Aloe vera* on fasting insulin levels warrants further investigation, given that research suggests that these herbal medicines could be important in the management of pre-T2D [26,27,28,29].

Importantly the therapeutic role of herbal medicines for the treatment of pre-T2D and T2D, as has been suggested for metformin [30], may be dependent on gut microbiota abundance and diversity. Certainly, this may constitute an important methodological issue to further consider in future clinical studies. Recently it has been confirmed that the that the intestinal microbiome in individuals diagnosed with preT2D or T2D presents a complex clinical picture with the host that will require consideration [31].

## 5. Conclusions

A diagnosis of pre-T2D describes a state of hyperglycemia that is outside of the normal clinical boundary and, as such, does not meet the criteria for a diagnosis of T2D. Non-pharmaceutical products for the management of chronic diseases present an attractive alternative, especially when patients present as symptomless. Clinical investigations with subjects presenting with pre-T2D can be difficult to conduct and can lead to significant study limitations. The randomized, double-blinded design was utilized so as to gain an appreciation of the safety and efficacy of this polyherbal natural formulation on changes in markers of glycemic function over 12 weeks, as measured by validated laboratory assessments. Notwithstanding, efficacy was not confirmed largely due to the short study duration and a wide initial FPG range.

This study included subjects with low levels of IFG and, consequently, a further limitation of this study was that some subjects were incorrectly included in this study. Further, once they were removed, this study was insufficiently powered to detect a difference of 10% between study arms. The challenges associated with a diagnosis of pre-T2D should include lifestyle factors such as subjects not adhering to prudent nutritional and physical activity lifestyles.

## Figures and Tables

**Figure 1 medicines-07-00022-f001:**
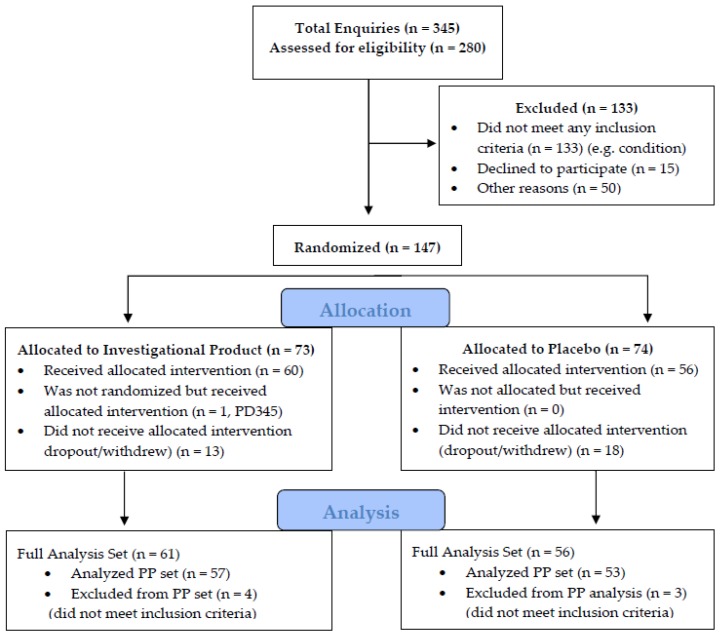
CONsolidated Standards of Reporting Trials (CONSORT) flow diagram of subjects relevant to enquiries—enrolments and randomization.

**Figure 2 medicines-07-00022-f002:**
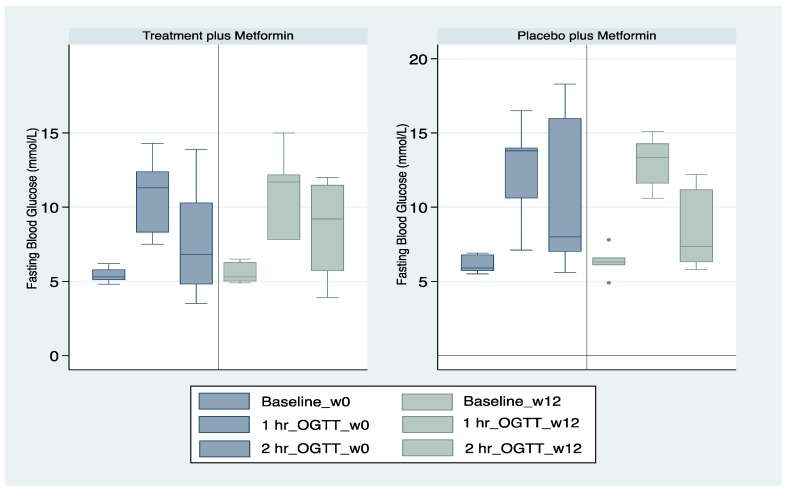
Subjects administered metformin with the investigational formulation or placebo.

**Figure 3 medicines-07-00022-f003:**
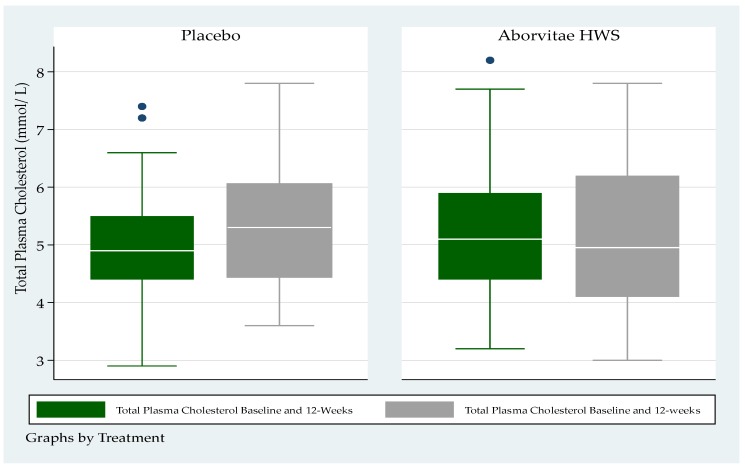
Total plasma cholesterol (mmol/L) at baseline and at the end of the study (week 12) (*p* > 0.05).

**Figure 4 medicines-07-00022-f004:**
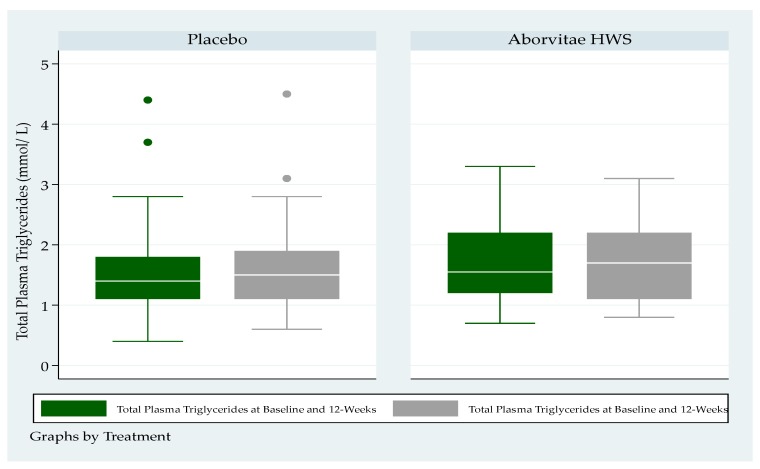
Total plasma triglycerides (mmol/L) at baseline and at the end of the study (12-weeks) (*p* > 0.05).

**Table 1 medicines-07-00022-t001:** Baseline characteristics (full analysis set population).

	Intervention Arm	Placebo Arm	*p*–Value
FACTORS	n = 61	n = 56	
Sex			
M [%] (n): F [%] (n)	57% (35): 43% (26)	48% (27): 52% (29)	-
Age [mean (SD years)			
M: F	61.1 (12.6): 60.8 (8.6)	63.7 (11.4): 60.1 (9.0)	-
Height [mean (SD) cm]	169.6 (10.8)	168.5 (10.5)	0.55
Weight [mean (SD) kg]	90.7 (19.1)	90.0 (18.9)	0.84
BMI ≥ 25 [%]	89% (54)	95% (53)	0.68
FPG [5.5–6.9 mmol/L] [%]	66% (40)	62% (35)	0.73
FPG, mean (SD)	5.9 (0.9) mmol/L	5.9 (0.8) mmol/L	0.94
OGTT 1 h ≥ 8.6 mmol/L [%]	93% (57)	95% (53)	-
1 h post OGTT mean (SD)	11.3 (2.4) mmol/L	10.7 (3.1) mmol/L	0.25
2 h post OGTT mean (SD)	8.2 (3.1) mmol/L	7.8 (3.0) mmol/L	0.49
Insulin mU/L fast mean (SD)	16.6 (8.1)	17.6 (11.4)	0.60
Insulin mU/L 1 h mean (SD)	130.0 (76.7)	130.9 (71.9)	0.95
Insulin mU/L 2 h mean (SD)	101.4 (73.8)	102.7 (71.5)	0.93
HbA1c % fast mean (SD)	5.7 (0.4)	5.6 (0.5)	0.25
With prescribed metformin FBG mmol/L 1 h post OGTT mean (SD) 2 h post OGTT mean (SD)	7/61 (11.5%) 6.2 (0.6) m12.9 (3.1) 10.8 (5.0)	10/56 (17.9%) 5.4 (0.5) 10.8 (3.1) 7.5 (3.4)	0.35

Notes: BMI = body mass index; FPG = fasting plasma glucose; OGTT = oral glucose tolerance test.

**Table 2 medicines-07-00022-t002:** Impaired fasting glucose (FAS population).

Proportion of Patients with IFG at Baseline and Week 12 (Screened Population)			
	Test Arm (n) % n = 61	Placebo Arm (n) % n = 56	*p* Value
**Baseline IFG**			
Yes	42 (69)	40 (71)	0.76
No	19 (31)	16 (29)	
**Week 12 IFG**			
Yes	40 (71)	35 (69)	0.75
No	16 (29)	16 (31)	
**Proportion of Patients with IFG at Baseline and Week 12** **(Population Meeting Inclusion/Exclusion Criteria)**			
**Baseline IFG**			
Yes	20 (35)	20 (38)	0.77
No	37 (65)	33 (62)	
**Week 12 IFG**			
Yes	39 (75)	33 (69)	0.49
No	13 (25)	15 (31)	
**Proportion of Patients with IFG at Week 12 Who Had IFG at Baseline** **(Screened Population)**			
**Baseline IFG**			
	**40**	**35**	
**Week 12 IFG**			
Yes	14 (39)	13 (43)	
No	22 (61)	17 (57)	0.71
**Proportion of Patients with IFG at Week 12 Who Had IFG at Baseline** **(Population Meeting Inclusion/Exclusion Criteria)**			
**Baseline IFG**			
	**37**	**33**	
Yes	11 (33)	12 (43)	
No	22 (67)	16 (57)	0.44

**Table 3 medicines-07-00022-t003:** Impaired glucose tolerance for the FAS population and PP population.

**Proportion of Patients with IGT at Baseline and Week 12 (FAS Population)**				
		**Test Arm (n) %**	**Placebo Arm (n) %**	***p* Value**
**Baseline IGT**		**n = 61**	**n = 56**	
Yes		23 (38)	16 (29)	
No		37 (62)	40 (71)	0.27
**Week 12 IGT**				
Yes		15 (27)	11 (22)	
No		41 (73)	40 (78)	0.53
**Proportion of Patients with IGT at Baseline and Week 12 (PP Population)**				
**Baseline IGT**		**n = 57**	**n = 53**	
Yes		20 (35)	15 (28)	
No		36 (64)	38 (72)	0.41
**Week 12 IGT**				
Yes		13 (25)	11 (23)	
No		39 (75)	37 (77)	0.81
**Proportion of Patients with IFG at Week 12 Who had IGT at Baseline (FAS Population)**				
**Baseline IFG**				
Yes		23	16	
**Week 12 IFG**				
Yes		12 (52)	8 (53)	
No		11 (48)	7 (47)	0.94
**Proportion of Patients with IGT at Week 12 Who had IGT at Baseline (PP Population)**				
**Baseline IFG**				
		20	15	
Yes		10 (50)	8 (57)	
No		10 (50)	6 (43)	0.68

**Table 4 medicines-07-00022-t004:** Changes in HbA1c (FAS population).

**FAS Population**	**Intervention Arm**	**Placebo Arm**	***p* Value**
	**n = 61**	**n = 56**	
Baseline HbA1c % (mean SD)	7.7 (0.4)	5.6 (0.5)	0.25
Week 12 HbA1c % (mean SD)	5.8 (0.4)	5.6 (0.6)	0.15
Change in HbA1c % (mean SD)	0.0 (0.2)	0.0 (0.3)	0.88
**PP population**	**Intervention Arm**	**Placebo Arm**	***p* value**
	**n = 57**	**n = 53**	
Baseline HbA1c % (mean SD)	5.7 (0.4)	5.6 (0.5)	0.24
Week 12 HbA1c % (mean SD)	5.7 (0.4)	5.6 (0.6)	0.23
Change in HbA1c % (mean SD)	0.0 (0.2)	0.0 (0.3)	0.46

**Table 5 medicines-07-00022-t005:** SF12 at baseline and at 12 weeks.

**FAS Population**	**Intervention Arm**	**Placebo Arm**	***p* Value**
	**n = 61**	**n = 56**	
SF12 baseline physical score (mean SD)	47.2 (9.0)	47.0 (8.4)	0.93
SF12 baseline mental score (mean SD)	50.5 (11.6)	51.0 (9.3)	0.86
SF12 week 12 physical score (mean SD)	46.1 (8.5)	47.1 (8.5)	0.60
SF12 week 12 mental score (mean SD)	50.4 (10.1)	53.2 (8.2)	0.20
Change in SF12 physical score baseline to week 12, mean (SD)	–0.2 (9.0)	0.1 (7.8)	0.88
Change in SF12 mental score baseline to week 12, mean (SD)	0.3 (9.5)	1.2 (9.2)	0.69
**PP Population**	**Intervention Arm**	**Placebo Arm**	***p* value**
	**n = 57**	**n = 53**	
SF12 baseline physical score (mean SD)	47.3 (9.1)	47.3 (8.0)	0.99
SF12 baseline mental score (mean SD)	51.1 (11.6)	50.7 (9.3)	0.86
SF12 week 12 physical score (mean SD)	46.3 (8.5)	47.2 (8.8)	0.67
SF12 week 12 mental score (mean SD)	50.0 (10.2)	53.2 (8.4)	0.91
Change in SF12 physical score baseline to week 12, mean (SD)	–0.4 (9.1)	–0.2 (7.6)	0.91
Change in SF12 mental score baseline to week 12, mean (SD)	–0.9 (7.9)	1.4 (9.6)	0.30

**Table 6 medicines-07-00022-t006:** Additional physiological measures.

Factor	Intervention Arm	Control Arm	*p* Value
**FAS population**	**n = 61**	**n = 56**	
BMI baseline (mean SD)	31.4 (6.3)	31.8 (5.5)	0.68
BMI week 12 (mean SD)	31.2 (5.2)	32.4 (9.0)	0.51
Fasting plasma glucose baseline (mean SD)	5.9 (0.9)	5.9 (0.8)	0.96
OGTT baseline 1 h (mean SD)	11.3 (2.4)	10.7 (3.1)	0.25
OGTT baseline 2 h (mean SD)	8.2 (3.1)	7.8 (3.0)	0.49
Fasting plasma glucose week 12 mean (SD)	5.8 (0.7)	6.0 (0.9)	0.21
1 hr post OGTT week 12 (mean SD)	11.1 (2.8)	10.4 (3.3)	0.20
2 hr post OGTT week 12 (mean SD)	7.6 (2.9)	7.9 (3.1)	0.64
Insulin levels mU/L baseline fasting (mean SD)	16.6 (8.1)	17.6 (11.4)	0.60
Insulin levels mU/L 1 h (mean SD)	130.0 (76.7)	130.9 (71.9)	0.95
Insulin levels mU/L 2 h (mean SD)	101.4 (73.8)	102.7 (72.5)	0.93
Insulin levels mU/L week 12 fasting (mean SD)	15.7 (9.3)	18.0 (9.8)	0.21
Insulin levels mU/L week 12 1 h (mean SD)	121.7 (74.1)	120.5 (67.0)	0.56
Insulin levels mU/L week 12 2 h (mean SD)	97.5 (82.9)	101.2 (67.6)	0.82
**PP population**	**n = 57**	**n = 53**	
BMI baseline (mean SD)	31.9 (6.2)	32.3 (5.2)	0.77
BMI week 12 (mean SD)	31.5 (5.0)	33.2 (8.7)	0.32
Fasting plasma glucose baseline (mean SD)	5.9 (0.9)	5.9 (0.9)	0.94
OGTT baseline 1 h (mean SD)	11.3 (2.5)	10.8 (3.0)	0.30
OGTT baseline 2 h (mean SD)	8.0 (3.1)	7.7 (3.0)	0.62
Fasting plasma glucose week 12 mean (SD)	5.7 (0.7)	6.0 (0.9)	0.14
1 hr post OGTT week 12 (mean SD)	11.0 (2.8)	10.2 (3.4)	0.23
2 hr post OGTT week 12 (mean SD)	7.3 (2.7)	7.7 (3.1)	0.50
Insulin levels mU/L baseline fasting (mean SD)	16.9 (8.2)	18.1 (11.5)	0.56
Insulin levels mU/L 1 h (mean SD)	134.1 (74.8)	134.7 (71.0)	0.97
Insulin levels mU/L 2 h (mean SD)	99.9 (72.1)	104.9 (72.2)	0.74
Insulin levels mU/L week 12 fasting (mean SD)	16.1 (9.5)	18.6 (9.9)	0.20
Insulin levels mU/L week 12 1 h (mean SD)	120.8 (73.2)	133.5 (67.1)	0.41
Insulin levels mU/L week 12 2 h (mean SD)	90.2 (78.4)	103.0 (69.6)	0.44

**Table 7 medicines-07-00022-t007:** Subgroup analysis of primary endpoint for participants with an FPG level in the range of 6.1–6.9 mmol/L.

Factor	Intervention Arm	Placebo Arm	*p* Value
	**n = 16**	**n = 15**	
**Gender** **M** **F**	12 4	8 7	0.21
BMI (mean SD) BMI ≥ 25 yes	29.2 (4.0) 15 of 16	35.1 (4.7) 15 of 15	<0.001 0.32
Height cm (mean SD)	171.1 (9.1)	168.3 (10.0)	0.42
Weight kg (mean SD)	88.4 (15.5)	100.2 (17.3)	0.05
Baseline blood glucose (mean SD)	6.5 (0.2)	6.5 (0.3)	0.89
Blood glucose 1 h post OGTT (mean SD)	12.3 (2.5)	12.7 (2.2)	0.67
Blood glucose 2 h post OGTT (mean SD)	10.1 (3.6)	9.0 (2.7)	0.35
Insulin levels mU/L baseline fasting (mean SD)	16.1 (8.5)	27.1 (16.1)	0.022
Insulin levels mU/L baseline 1 h (mean SD)	103.9 (47.2)	167.0 (82.5)	0.022
Insulin levels mU/L baseline 2 h (mean SD)	110.5 (58.6)	158.2 (88.3)	0.10
Baseline HbA1c% (mean SD)	5.9 (0.6)	5.9 (0.3)	0.65

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
