# Peer review of "A Double-Blind Randomized Placebo-Controlled Study Assessing the Safety, Tolerability and Efficacy of a Herbal Medicine Containing Pycnogenol Combined with Papain and Aloe vera in the Prevention and Management of Pre-Diabetes"

_medicines, 2020, doi:10.3390/medicines7040022_

Round 1
Reviewer 1 Report
The paper is fairly well written
The introduction can benefit from a rationale for including papain and aloe vera
line 106: I believe "women" is missing
Line 140, 144, 154: the formulation includes pycogrnol and papain only but not aloe vera. Is this a different formulation?
Author Response
We thank the reviewer for the queries and provide the following replies and as such have amended the manuscript accordingly.
- In the introduction we have added text relevant to the inclusion of Carica papaya and Aloe vera in the formulation investigated with appropriate references.
- Line 106 has been amended.
- Lines 140, 144 and 154...have been rectified to include Aloe vera.
Reviewer 2 Report
Review of
A Double-Blind Randomized Placebo-Controlled 2 Study Assessing the Safety, Tolerability and Efficacy 3 of an Herbal Medicine Containing Pycnogenol 4 Combined with Papain and Aloe vera in the 5 Prevention and Management of Pre-Diabetes.
- Title and Abstract reflect precisely the main purpose of the manuscript. The study is interesting, and the authors tried to contribute to this field of research
- Abstract: The authors missed to list the full name of the IFG before the first abbreviation. The authors need to rephrase the last sentence, because in this manuscript no studies have been done about the intestinal microbiome. It is stated in the Instructions for authors, https://www.mdpi.com/journal/medicines/instructions#preparation, that the total length of the abstract should be a maximum of about 200 The authors need to rephrase the abstract in order for it to be shorter. Currently the Abstract consists of 264 words.
- In the Introduction more data should be presented on similar studies. When describing the used herbal formulation containing pycnogenol, papain and Aloe vera, the latin name of the origin plants and the utilized plant organs must be presented.
- Materials and methods: in 2.2. Design of the Study – it is not clear why the authors mentioned that a total of four study visits took place, meanwhile, in the Results and the Discussions sections, they only presented the results for the Baseline and Week 12 of the study.
In 2.3. Outcome Measures – the authors described the used herbal formulation, containing two active principles, pycnogenol and papain. At the same time, in the Abstract and in the Introduction sections, they described a herbal formulation containing pycnogenol, papain and Aloe vera. The latin name of the origin plants, the utilized plant organs and the source of the used herbal supplement or the conditions for obtaining it must be presented. The proportion of the combination and the administered doses must be described here, rather than the Abstract.
The authors need to detail the differences between FAS population and PP population if it is relevant to the study.
In 2.3. Outcome Measures and in 2.5. Statistical Analysis there are several sentences that are overlapping with this bibliographic source: https://journals.plos.org/plosone/article?id=10.1371/journal.pone.0138646). The authors need to rephrase the sentences and include this bibliographic source in the list of references.
- Results: The text between line 257 and 259 appears one more time between line 260 and 262. In Figure 3 and Figure 4 the obtained values must appear.
- Discussions: The authors need to discuss all the presented results and the correlation between them.
Because no studies were conducted, in relation to the intestinal microbiome in this manuscript, in my opinion, the discussions about this topic are not relevant.
The authors need to complete the Discussions section with the comparison of the obtained data with the similar data obtained in other similar studies. Future research directions may also be mentioned.
- References: The authors need to check the list of references and correct it according to the Instructions for authors. The authors need to include this bibliographic source: https://journals.plos.org/plosone/article?id=10.1371/journal.pone.0138646) in the list of references

Author Response
We thank the reviewer for the comments that, we think very much enhance the submission. We provide the following answers to the queries.
- We have reduced the word count in the abstract. Abbreviation amended.
- Re the introduction...we have amended the active components with the addition of Latin names and have added similar references as suggested.
- We have clarified in the methods why subjects attended 4 visits and only at baseline and at 12 weeks were pathology tests requested.
- Outcomes 2.3: we have amended accordingly the formulation and clarified the individual concentrations of the actives in the vial.
- We have clarified the difference between FAS and PP.
- In outcomes and statistical section 2.3: amended as advised and reference included.
- In results section duplicate text removed.
- Figure 3 and 4 values added as suggested.
- Additional references have included in the discussion as suggested. Moreover, we have further discussed the results. And the suggested reference has been added to the list.
- We felt that it was appropriate to leave in the discussion a brief comment re the intestinal microbiome given that in part a discussion could include this...notwithstanding though rightfully we have removed any such microbiome text from the abstract.